## Comment

behaviour, ecology, theoretical biology

ideal free distribution, animal personality, individual-based modelling, optimal foraging

**Authors for correspondence:**
Christoph Netz
e-mail: c.f.g.netz@rug.nl
Franz J. Weissing
e-mail: f.j.weissing@rug.nl

# Details matter when modelling the effects of animal personality on the spatial distribution of foragers

Christoph Netz, Aparajitha Ramesh, Jakob Gismann, Pratik R. Gupte and Franz J. Weissing

Groningen Institute for Evolutionary Life Sciences, University of Groningen, Groningen, The Netherlands

CN, 0000-0001-9309-8964; AR, 0000-0002-7200-1366; JG, 0000-0002-2570-590X; PRG, 0000-0001-5294-7819; FJW, 0000-0003-3281-663X

By means of a simulation study, DiNuzzo & Griffen [1] investigate whether individual variation in a personality trait can explain 'undermatching', an often-observed deviation from the ideal free distribution (IFD). Here, we raise five points of concern about this study, regarding (i) the interpretation of the results in terms of personality variation; (ii) deficiencies in the technical implementation of the model, leading to wrong conclusions; (iii) the effects of population size on deviations from the IFD; (iv) the measure used for quantifying deviations from the IFD and (v) the analysis of the mud crab data. Finally, we provide an overview of the evolutionary ramifications of the relation between animal personality and the IFD.

## 1. Personality variation and the IFD

The individuals in DiNuzzo & Griffen's model tend to maximize their intake rate. At each point in time, they are perfectly informed about the distribution of resources (which remains constant) and the distribution of foragers (which can change due to movement). Individuals differ in 'activity', that is the rate at which they recognize that their current intake rate is suboptimal; once they observe a discrepancy, they move instantaneously to the habitat patch yielding a maximal intake rate. In this model, each individual has to move at most once: if all individuals have moved (or stayed at their initial position, as this already yielded a maximal intake rate), the IFD is reached. It is therefore obvious that less active individuals that, by definition, take on average more time steps for making a movement decision, retard the approach of the population to the IFD. Hence, it is also obvious that the 'time to reach IFD' increases with an increase of the proportion of inactive individuals. In other words, it is not personality variation *per se* that retards the approach to the IFD but rather the presence of inefficient movers.

## 2. Problems with the technical implementation of the model

Above we argued that it is obvious that the 'time to reach IFD' increases with the proportion of inactive individuals. In view of this, it is surprising that DiNuzzo & Griffen report a hump-shaped relationship in one of their simulation scenarios (their fig. 4e) and even a monotonic decline in the time to reach IFD with increasing proportions of inactive individuals in case of a type II functional response (their electronic supplementary material, fig. S1, reproduced here in figure 1a). We think both results are artefacts. The pattern in their electronic supplementary material fig. S1 is caused by a comparison between intake rates calculated with two different formulas. As a consequence, individuals can 'believe' that they are already in a habitat maximizing their intake rate, while really they are not.

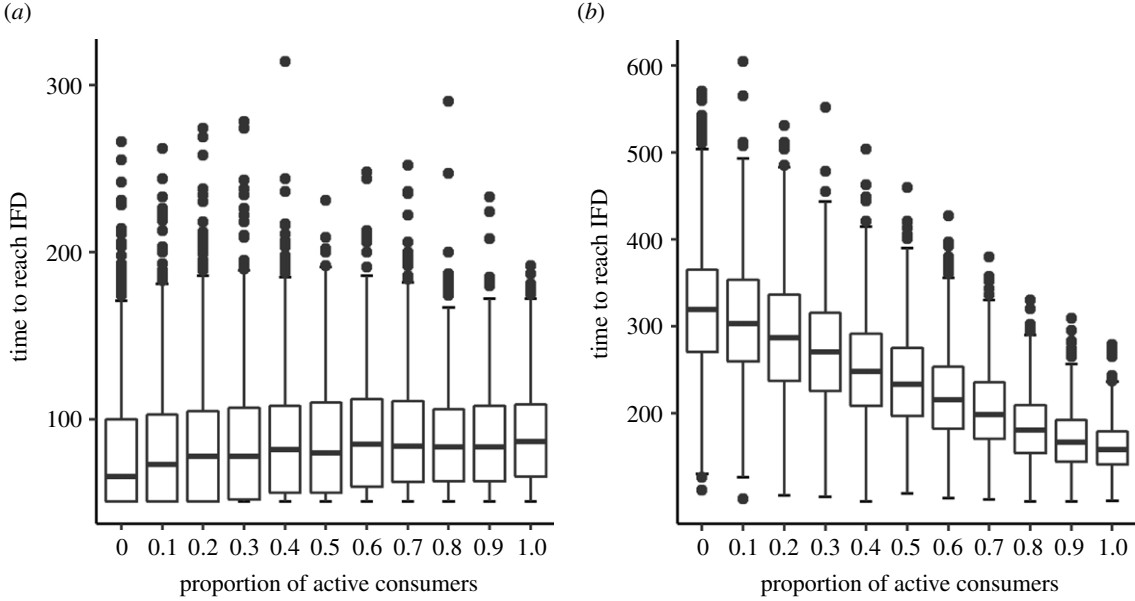

**Figure 1.** Replication of DiNuzzo & Griffen's electronic supplementary material fig. S1 (*a*) using their original NetLogo code and (*b*) using a corrected version of their code. Both panels show the time to reach the ideal free distribution (IFD) for various proportions of 'active' (80% activity) and 'inactive' (20% activity) consumers with a type II functional response in 1000 replicate simulations. According to DiNuzzo & Griffen's NetLogo code, the time-to-IFD increases with the proportion of active consumers. A corrected version of the code (see our electronic supplementary material [3] for details) yields the expected pattern of decreasing waiting times with increasing proportions of active consumers.

In addition, an incorrect formula of a ratio-dependent functional response type 2 is used (following [2]). A detailed explanation of these mistakes can be found in our electronic supplementary material [3]. If these mistakes are corrected, the time to reach IFD shows the expected increasing trend with the proportion of inactive individuals (figure 1*b*), rather than the decreasing trend reported by DiNuzzo & Griffen. Hence, a saturating type II functional response leads to a similar relationship between the proportion of active consumers and time-to-IFD as an unlimited linear (type I) functional response. Special explanations for discrepancies between type I and type II models (the 'domino effect' explanation in electronic supplementary material, 1.4 of [1]) are not needed and are actually misleading.

We can further show by a simple mathematical argument that the correspondence between the two model variants considered by DiNuzzo & Griffen should be even stronger: the special version of the type II functional response used by DiNuzzo & Griffen (following [2]) should lead to *exactly* the same time-to-IFD and the same consumer distribution over patches as their type I functional response (see part 3 of our electronic supplementary material [3]). We were therefore surprised that our figure 1*b* does not exactly match with fig. 3 in [1]: it generally takes 100 time steps longer to reach the IFD. Rerunning the scenario underlying fig. 3 in [1] with DiNuzzo & Griffen's published NetLogo code, we did obtain an exact replicate of our figure 1*b*. We conclude that DiNuzzo & Griffen must have used a different version of their simulation program to produce their fig. 3.

In addition, the simulation program in [1] produces a substantial bias in reported time to reach the IFD. Each simulation run stops once movement has ceased for 50 time steps, assuming that this is a clear indication that the IFD has been reached. The problem is that movement can cease for 50 time steps even in situations where the population is still far from an IFD (figure 2*a*). This easily happens in populations with a large proportion of highly inactive individuals:

the lack of movement may just reflect the reluctance of these individuals to move (rather than having reached a habitat with maximal intake rate, where movement is no longer necessary). Figure 2 shows two replications of fig. 4*e* in [1], one with the published NetLogo code (figure 2*b*) and a second with an improved version (see our electronic supplementary material [3]) where DiNuzzo & Griffen's stopping criterion is replaced by a check whether the IFD has indeed been reached (figure 2*c*). It is obvious that the stopping criterion has a large effect on the simulation outcome. Notice that neither outcome shows the puzzling 'hump' of fig. 4*e* in [1]. As we produced figure 2*b* with DiNuzzo & Griffen's published NetLogo code, we have to conclude again that a different version of their simulation program was used to derive their fig. 4*e*.

A more detailed account of the technical issues reported above (and some additional issues) and corrected versions of the NetLogo program can be found in our electronic supplementary material [3].

## 3. Effects of population size

DiNuzzo & Griffen investigated the effect of population size on the time to reach the IFD. However, the timescale of their model implementation is quite different from a 'natural' timescale. In their simulation program, individuals make decisions sequentially, and only one individual can make a decision in each time step. As in a larger population more individuals have to take decisions, this automatically increases the time to reach a certain target distribution. Moreover, the time to reach the IFD is inflated by the fact that active individuals are restricted in their movement because they have to 'wait' for inactive individuals. For these reasons, it is more natural to use a continuous timescale, where individuals take movement decisions independently of each other, at a rate that is proportional to their activity level. This can be done in a straightforward manner, by translating the discrete-time model of DiNuzzo & Griffen into an

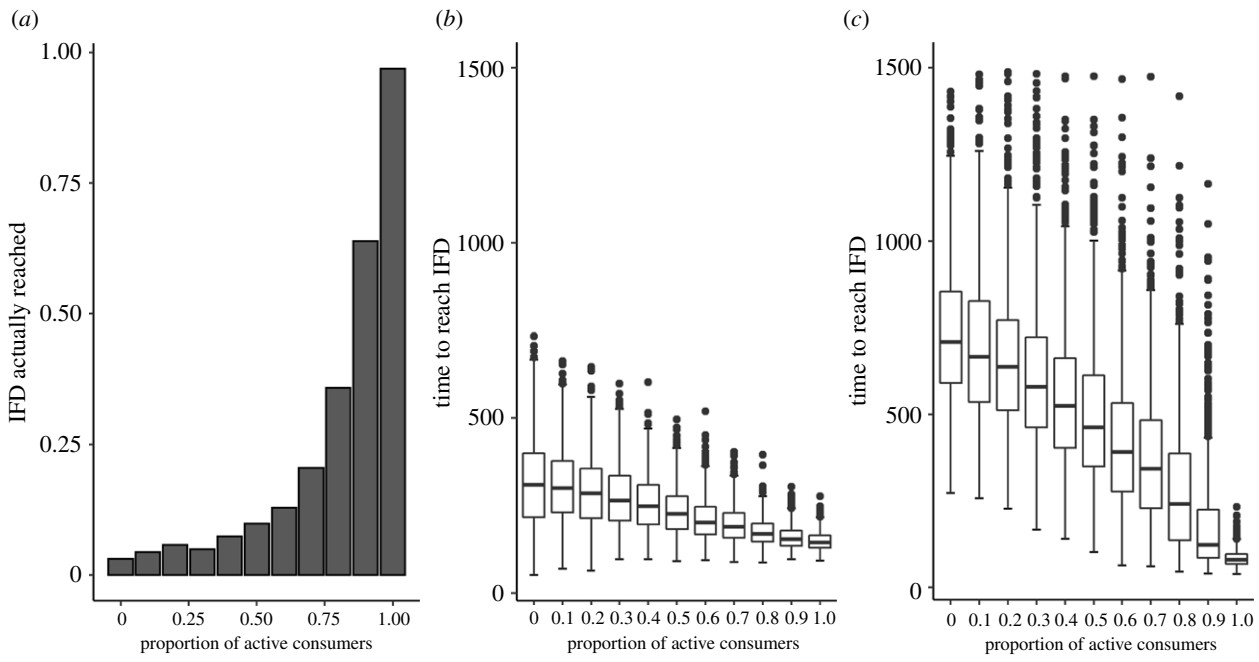

**Figure 2.** Systematic bias in outcomes due to premature termination of simulations. The NetLogo code underlying the simulations in [1] assumes that the IFD is reached after 50 time steps of inactivity. (*a*) The proportion of simulations that have actually reached the IFD after 50 time steps of inactivity in the scenario underlying fig. 4*e* in [1]. (*b*) Replication of fig. 4*e*, using DiNuzzo & Griffen's NetLogo code. (*c*) The same set of simulations for an improved version of the NetLogo code, where a simulation now stops when the IFD is actually reached. In all simulations, 'active' consumers have an activity level of 90% while 'inactive' consumers have an activity level of 10%.

otherwise equivalent event-based model (making use of the Gillespie algorithm [4]; a description and implementation of such a model can be found in [5]). Figure 3 shows how in the event-based version of the model the time to reach the IFD depends on the population size $N$ and the proportion of active individuals. For each population size, the time to reach the IFD is, as expected, positively related to the proportion of inactive individuals. However, the event-based version of the model does not support DiNuzzo & Griffen's conclusion that the time to reach the IFD increases with population size. This only occurs for very low population densities ($N = 8$ and $N = 40$ in figure 3), and even in these cases, the effect is small. At higher population sizes, the time to reach the IFD *decreases* with population size: as shown in figure 3, the IFD is reached much faster in a population with $N = 1000$ individuals than in any of the smaller populations. This can be explained as follows. In the case of the low population sizes considered by DiNuzzo & Griffen, the initial density of individuals is very low (typically only one individual per patch). In such a case, an individual can only improve its intake rate by moving to a more profitable patch. In case of a large population size (and a higher initial density per patch), there is an additional option: if an individual on a patch decides to leave in order to improve its intake rate elsewhere, all remaining individuals on that patch profit as their intake rate increases due to alleviated within-patch competition (see [6]). This effect is not addressed by the study of DiNuzzo & Griffen, although the authors state, 'in most natural systems, there are many more consumers than patches'.

while low-resource patches are relatively overexploited. Yet, they devote only one figure (their fig. 2) to this phenomenon. In general, they quantify deviations from the IFD by measuring the time to reach the IFD. This measure has at least three disadvantages. First, 'time-to-IFD' is determined by the last individual that moves to a patch with an optimal intake rate. In other words, a single individual with very low activity can have a very large effect on the time-to-IFD. Second, 'time-to-IFD' depends on the initial conditions; it takes longer to reach the IFD if the initial spatial distribution of individuals differs a lot from the IFD. Third, 'time-to-IFD' is only a sensible measure when the IFD is actually reached. This, however, will only be the case in highly standardized simulation models with a fixed resource distribution. As stated by DiNuzzo & Griffen: 'In most systems, the IFD is a moving target owing to temporal environmental variation and directional change (i.e. habitat degradation)'. In §1.5 of their electronic supplementary material, DiNuzzo & Griffen show some simulation results for a scenario with temporally varying patch quality. Surprisingly, 'time-to-IFD' is also used for this scenario (their electronic supplementary material fig. S2), where it is difficult for us to understand how the IFD can ever be reached in the case of rapid environmental change. How can movement cease for 50 time steps (the criterion for reaching the IFD) if the distribution of patch qualities changes completely every 10 or 20 time steps? Under such changing conditions, we would advocate using a more robust, population-level measure for deviations from the IFD, such as the variance in intake rates across patches.

## 4. Quantifying the approach to the IFD

DiNuzzo & Griffen conducted their study in order to investigate whether personality differences can explain 'undermatching', the commonly observed phenomenon that high-resource patches tend to be relatively underexploited,

## 5. Analysis of the mud crab system

We are puzzled by the fact that DiNuzzo & Griffen revert to a simple calculation of activity ratios in their analysis of the

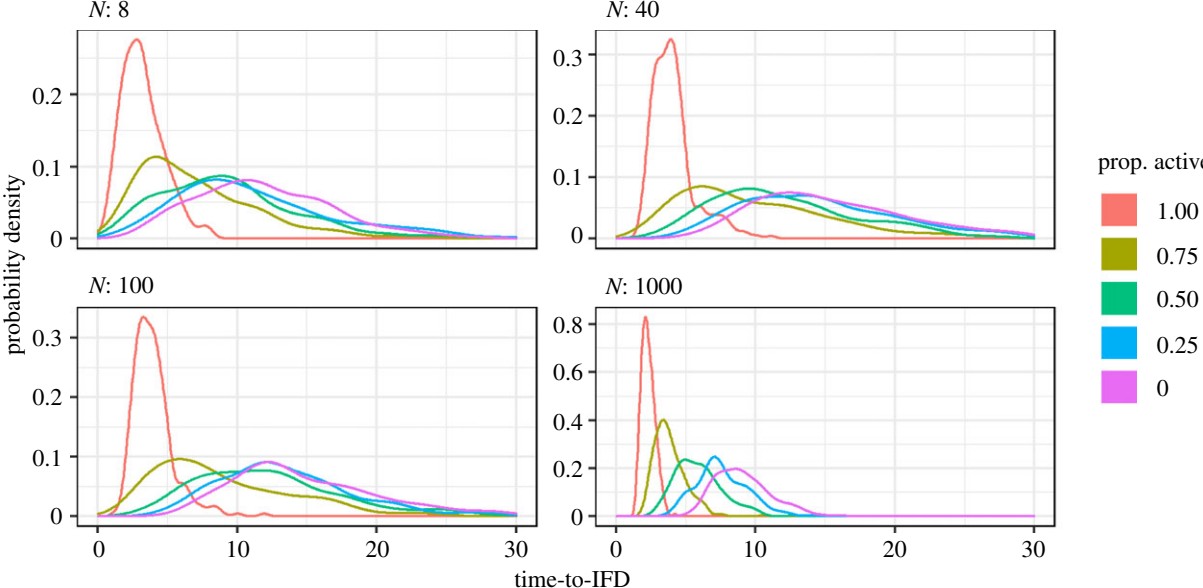

**Figure 3.** Probability distributions of the time until the ideal-free distribution is reached, based on 1000 replicate simulations per setting. In a system with 49 habitat patches, the panels show how the time to reach IFD depends on the proportion of 'active' (movement rate 0.8) and 'inactive' (movement rate 0.2) individuals for four population sizes N. (Online version in colour.)

refuge use data on the mud crab *Panopeus herbstii* [7] instead of taking advantage of their individual-based model. The model becomes necessary because such a simple calculation does not suffice, as it ignores the distribution of personality in the population. Hence, their fig. 5 illustrates the influence of personality on the IFD only in the sense that no single crab is 'ideal' in immediately leaving its refuge and moving to the patch with highest profitability, but not the implications of the distribution of activity levels in the population. Additionally, the data come from a special (predation cue) treatment, not from standard conditions, and the crabs differ substantially in size (actually body size is used as a proxy for activity level) and accordingly also in their resource needs and competitive abilities.

## 6. Outlook

We have the impression that DiNuzzo & Griffen view 'personalities' mainly as (maladaptive) deviations from optimal or efficient behaviour. By contrast, many studies show that personality variation is often shaped by adaptive evolution [8–14]. For example, Wolf *et al.* [6] demonstrate that 'inactivity' (called 'unresponsiveness' in [6]) may be viewed as an efficient strategy in achieving a high foraging success and approaching an IFD. An adaptive perspective on personality variation leads to novel eco-evolutionary questions regarding the interplay of individual behavioural variation and the spatial distribution of foragers. The IFD is a prototype example of a model linking ecology (the spatial distribution of foragers) to evolution (optimal or evolutionarily stable movement decisions). Future research is needed to reconcile the IFD with the eco-evolutionary causes and consequences of personality for at least two reasons: first, the IFD model

presupposes that the resource intake rate is a proxy for fitness [15]. But how, then, can different personality types persist at stable proportions, when inactive individuals consistently achieve a lower intake rate than their more active conspecifics? Second, a personality perspective may change what spatial distribution is optimal. In animals, differences in activity are usually associated with (adaptive) differences in energy metabolism [16]. When foraging individuals differ in energetic expenditure, they should not take maximizing the intake rate as their sole guiding principle [17]. In other words, individuals differing in activity should use different decision rules, and the optimal behaviour of a polymorphic population may, even at equilibrium, deviate considerably from the IFD of a monomorphic population.

**Data accessibility.** Our electronic supplementary material comprises (1) a technical analysis of DiNuzzo & Griffen's model code; (2) an altered version of their code used for figures 1 and 2; (3) a mathematical proof that the ratio-dependent functional responses of type I and II produce exactly the same movement rules; and (4) a concise description of our event-based simulation model [3]. The code for this model is deposited at Zenodo https://doi.org/10.5281/ZENODO.4537547 [5].

**Authors' contributions.** C.N., A.R., J.G., P.R.G. and F.J.W. conceived the contents of this comment. C.N. developed the simulation program. C.N. and F.J.W. wrote the article. All authors performed revisions and gave final approval of the paper.

All authors gave final approval for publication and agreed to be held accountable for the work performed therein.

**Competing interests.** We declare we have no competing interests.

**Funding.** We received funding from the European Research Council (ERC Advanced grant no. 789240) and the Netherlands Organisation for Scientific Research (Open Competition grant no. ALWOP.668).

**Acknowledgements.** We would like to thank B. van Beek and E. D. Schulte for bringing the equivalence of functional response type I and II to our attention, as well as three anonymous reviewers for their helpful comments on our manuscript.

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
