## [Peer Review File · Proceedings of the Royal Society B: Biological Sciences]

Review History

RSPB-2021-0367.R0 (Original submission)

Review form: Reviewer 1

Recommendation

Major revision is needed (please make suggestions in comments)

Scientific importance: Is the manuscript an original and important contribution to its field?

Acceptable

General interest: Is the paper of sufficient general interest?

Marginal

Quality of the paper: Is the overall quality of the paper suitable?

Acceptable

Is the length of the paper justified?

Yes

Should the paper be seen by a specialist statistical reviewer?

No

Do you have any concerns about statistical analyses in this paper? If so, please specify them explicitly in your report.

No

It is a condition of publication that authors make their supporting data, code and materials available - either as supplementary material or hosted in an external repository. Please rate, if applicable, the supporting data on the following criteria.

Is it accessible?

Yes

Is it clear?

Yes

Is it adequate?

No

Do you have any ethical concerns with this paper?

No

Comments to the Author

In the article, "Details matter when modelling the effects of animal personality on the spatial distribution of foragers" the authors provide a critique of a recent publication and model produced in DiNuzzo and Griffen (2020). The authors provide some good comments on how assumptions were built into the model, highlight some errors, and touch upon how results interpretation may change depending on the ecological context. However, there are a number of issues with this critique and points that seem to have been misunderstood in the original paper. For instance, the authors seem to not understand why a Type II functional response limiting max consumption rate was used and why it caused results patterns to reverse. Additionally, the authors state a number of times that changes to the model may alter findings, but frequently do not actually demonstrate how the findings would be altered. This manuscript should focus more on specifically describing and expanding upon how model differences affect ecological interpretation, and demonstrate with simulations how results would differ when coding/parameters are changed.

Specific comments below

Point 1

The authors make a good point that DiNuzzo and Griffen's model shows that the presence of inefficient movers is the underlying cause behind populations taking longer to reach the IFD in this model. However, the authors seem to be assuming that DiNuzzo and Griffen are advocating that variation in personality almost always slows down the approach to IFD when the paper is more a demonstration of how personalities CAN slow down the approach to IFD and provide a simple example. There are a number of reasons why individuals may not be equal in movement efficiency (e.g. injury, sickness, physiological differences). Similarly, there are a number of ways that individual personality can affect movement. The most apparent personality effect is that shy, docile individuals often spend more time in refuge and less time active, even when hungry in many circumstances. However, in a number of species, shy individuals often have smaller home ranges (e.g. Watson & Miller 1971; Spiegel et al. 2015), shorter dispersal distances (e.g. Cote et al. 2010), or slower reaction time to changing food availability (e.g. van Overveld and Matthysen 2010) than bold individuals which can preclude them from finding optimal patches. Differences in competitive ability are already well-known to be a cause for deviations from IFD, but differences in personality are also often the reason why individuals are not equally competitive. Although not discussed by DiNuzzo and Griffen, boldness has the potential to do the reverse

where exceptionally fast individuals may quickly find resources and help the population reach IFD sooner, particularly if they attract conspecifics or communicate (e.g. bees).

Thus, DiNuzzo and Griffen were not so much flawed in the interpretation of their model findings as the interpretation could be extended to include anything that causes individuals to be inefficient in their movement. Should reduce the criticism here, but continue to point out how it is the presence of inefficient movers and factors that govern movement that retard approach to IFD.

Cote, J., Clobert, J., Brodin, T., Fogarty, S. & Sih, A. (2010). Personality-dependent dispersal: characterization, ontogeny and consequences for spatially structured populations. *Philos. Trans. R. Soc. Lond. B. Biol. Sci.*, 365, 4065–76

Spiegel, O., Leu, S.T., Sih, A., Godfrey, S.S. & Bull, C.M. (2015). When the going gets tough: behavioural type-dependent space use in the sleepy lizard changes as the season dries. *Proc. R. Soc. B Biol. Sci.*, 282, 20151768

van Overveld, T. & Matthysen, E. (2010). Personality predicts spatial responses to food manipulations in free-ranging great tits (*Parus major*). *Biol. Lett.*, 6, 187–90

Watson, B.Y.A. & Miller, G.R. (1971). Territory Size and Aggression in a Fluctuating Red Grouse Population. *J. Anim. Ecol.*, 40, 367–383

First point of the third paragraph: The authors question why different activity levels can stably coexist within a system, but that seems to be beyond the scope of the study for DiNuzzo and Griffen which was to show how personality variation can be responsible for IFD undermatching. There is an entire section of literature on why personalities may persist within a population. The quick answer is that all models are simplifications of complex systems and there are outside factors not considered here such as predation rate of different individuals (shy individuals have higher survival probability than bold, but bold can reproduce more because they gain more resources). Sih et al. (2015) provides a good overview on a number of potential mechanisms that may lead to populations maintaining multiple personality types across either short or long time scales (lifetime to multigenerational).

The second point of this third paragraph is a good one and ties into the first point. The authors should expand upon this idea that equilibrium states vary depending on diversity of individuals within a population and other external factors beyond resource availability. Providing specific examples, scenarios, and/or illustrations and models quantifying how much these equilibrium states should change depending on these factors would be valuable here.

Sih, A., Mathot, K. J., Moirón, M., Montiglio, P. O., Wolf, M., & Dingemanse, N. J. (2015). Animal personality and state-behaviour feedbacks: a review and guide for empiricists. *Trends in Ecology & Evolution*, 30(1), 50-60.

Point 2

In this section the authors outline some of their issues with the coding of the model and refer to a citation written by them and posted to github providing specific details on their model issues (citation 8). The technical notes of this citation need to be provided as supplementary material for this article and a url to the overall work should also be provided in the citation.

In these technical notes, the authors note some important points in the code of DiNuzzo and Griffen, but also themselves seem to have had some misunderstandings which are outlined below.

Note 2: “set quality (2 + random 8)” would have quality values go from 2 – 10 with 9 distinct levels of habitat quality, not 2 – 9 as the authors state.

Note 3: The authors make a great point here that designating 50 time steps as a stopping point may cause the simulation to end prematurely before IFD is reached and that this may be the reason for the hump in depicted in Figure 4e of DiNuzzo and Griffen. The authors state in their manuscript that in their model they no longer see this hump, but they do not provide a figure of this since Figure 1 of their paper does not ever adjust the personality types to be 0.9 and 0.1 rather 0.8 and 0.2.

Note 6: Here, the authors are concerned that there is an upper limit to the max feeding rate, set at 1, and that this causes a “bug” in the system where individuals choose not to move to patches with higher number of resources because they are on a patch that maxes their resource consumption limit, even though it has fewer overall resources than the richest available patch. This constitutes a major point of the authors’ submitted article, but the authors do not seem to have understood the reason behind also running the model using a Type II functional response in addition to simulations with a Type I functional response. Type I functional responses are rare in nature because they assume that individuals are never satiated and will consume whatever resources are available. However, eventually individuals’ needs are met and they stop consuming resources even when more are available in the region. Under these scenarios, there is no point for individuals to expend energy traveling to another location with more resources, potentially risking higher predation rates during travel, if they can already acquire all the resources they need in the present location. It is therefore more realistic to examine these scenarios using a type II functional response where individual feeding rate is capped and where individuals stop moving on patches that meet their needs but are not the richest in available resources. However, the increase in time to reach IFD with more active individuals is not actually so much a result of individuals taking more time to find the richest available patch because they are satiated on “suboptimal patches”, but because of changes to how patch resources are divided among individuals. Resources on a patch are no longer divided into ever smaller infinitesimal fractions as crabs are added (where a patch with 8 resources could theoretically support 4 crabs each consuming 2 resources or 400 crabs each consuming 0.02 resources, which is unrealistic), but instead resources are subtracted as crabs are added so eventually no more crabs can go on a certain patch because all resources would already be consumed (a patch with 8 resources could support 8 crabs consuming 1 resource; code lines 80 and 98 versus 89 and 107). This means that adding a crab to a patch has a much larger effect on available resources and can cause a chain reaction in additional movement as a single crab is added to a patch that had few available resources remaining. As the number of active individuals increase, so too would the probability that a chain reaction in movement would occur.

This difference in how resources are divided among population members and its effect on findings should have been highlighted better by DiNuzzo and Griffen since both calculations are valid under different ecological circumstances and produce very different results. But framing the results of Figure S1 as an error or artefact does not seem appropriate. It could however lend itself well to a discussion on ecological context and methods of modeling that context. Especially, if the authors run a sensitivity analyses varying the max feeding limit and report subsequent results. If the authors want to frame this as an error then they need to show that the results become qualitatively different when the mistakes highlighted in points 4 and 5 are corrected and should provide new model code in supplemental.

In the supplemental material authors also need to provide the code used to produce the model results depicted in Figure 1 as well as highlight the associated changes they made to the original model code of DiNuzzo and Griffen.

Point 3

Again, code for Figure 1 would best be provided in supplemental. Note that DiNuzzo and Griffen used 1000 replicate simulations rather than the 300 used here.

Point 4

The authors are correct that time to reach IFD is an imperfect measure for quantifying such

deviations in a natural system. This method is really only suitable for a highly standardized model like the one utilized by DiNuzzo and Griffen that only implements 2 personality types and runs 1000 simulations/scenario to help identify the variation in results due to differences in initial starting conditions and probability that simulation random-float numbers hit movement thresholds. The authors should focus this segment more on alternative methods to measure differences from IFD that work better for natural populations or more sophisticated models. This section would benefit greatly if the authors implement these different measurements of deviation from IFD in their own models and compare the results. Does using variance in intake rates across patches actually lead to different conclusions?

Point 5

It is true that refuge use does not necessarily translate to activity level as individuals can still be moving around to some degree while within a refuge. However, the authors seem to have misinterpreted how DiNuzzo and Griffen relate refuge use (what they call activity level) to their calculation of movement propensity. Here, refuge use was not treated as a 1-1 relationship to movement propensity but instead relied upon a prior study conducted by Belgrad and Griffen (2018) that quantified the relationship between individual refuge use and movement propensity to determine how much crabs would move in the model. The major criticism here then should not be that DiNuzzo and Griffen used the wrong study system since there is a fair amount of literature demonstrating that the species exhibits persistent individual differences in various behaviors that correlate with a number of other important characteristics. Rather the major point of concern is the number of relationships estimated in assigning population movement propensities for the model and the extra level of uncertainty built into their calculations. Instead of using movement propensities measured from a large sample of the population, DiNuzzo and Griffen estimate individual refuge use propensity from a size distribution of the population and then use that estimation to estimate movement propensity of the population. The authors should instead focus their critique on this level of uncertainty or that DiNuzzo and Griffen needed to use a study system where personality distribution did not have to be estimated.

Figures 3 and 4 provide graphs of how the degree of personality variation can affect the time to reach IFD from theoretical populations. The purpose of Figure 5, does not appear to really be about illustrating how variation in personality can affect the time to IFD which they do with their theoretical populations. Figure 5 is a rough calculation of how much longer it should take a natural population to reach IFD given that individuals do not follow "ideal" movement probabilities. If one were to vary the population personality distributions for such a calculation, it would no longer be a real-world personality distribution. Is it natural for 90% of individuals to have an activity level of 0.9? It would have been better if the DiNuzzo and Griffen paper used multiple populations to provide that information, but that also seems to be the point of the first part of their manuscript. Studies almost never report the personality distribution of populations and there is a lack of data on this subject. Still, they did find 10 studies that do seem to report that information so they could have potentially used that data to improve their estimates further.

Belgrad, B. A., & Griffen, B. D. (2018). Personality interacts with habitat quality to govern individual mortality and dispersal patterns. *Ecol. Evol.*, 8, 7216-7227.

Review form: Reviewer 2 (J Fryxell)

Recommendation

Accept as is

Scientific importance: Is the manuscript an original and important contribution to its field?

Excellent

General interest: Is the paper of sufficient general interest?

Excellent

Quality of the paper: Is the overall quality of the paper suitable?

Excellent

Is the length of the paper justified?

Yes

Should the paper be seen by a specialist statistical reviewer?

No

Do you have any concerns about statistical analyses in this paper? If so, please specify them explicitly in your report.

No

It is a condition of publication that authors make their supporting data, code and materials available - either as supplementary material or hosted in an external repository. Please rate, if applicable, the supporting data on the following criteria.

Is it accessible?

Yes

Is it clear?

Yes

Is it adequate?

Yes

Do you have any ethical concerns with this paper?

No

Comments to the Author

This manuscript clarifies and enriches some potentially key processes first outlined by DiNuzzo and Griffen (2020) influencing the time required for hypothetical populations to converge on an ideal free distribution and the commonly-observed tendency for under-matching to occur as a result. The authors central thesis is that any delays in decision-making will automatically extend the time to equilibrium by definition. This seems a subtle, but important, point since other many factors (e.g. variation among individuals due to differences due to internal motivation, physiological state, perception, or cognition) could act in this way in addition to variation in personality syndrome.

The authors also clarify that a key component in any such model is whether decisions are made sequentially by individuals across the population or simultaneously. Changing this assumption changed several outcomes in important ways. Such considerations about discrete vs continuous representations of time can fundamentally alter outcomes in many models in ecology, including the most basic of density-dependent formulations. Given that we can rarely (perhaps never) be sure that either continuous or discrete formulations are truly correct, it is important to recognize this as a key biological constraint on all model formulations. The authors provision of code for IDF formulation according to simultaneous decision-making is a most useful contribution to the field.

I am less enthusiastic about the authors' complaint that DiNuzzo and Griffen's (2020) crab population represents a less than perfect example of the processes discussed in the theoretical part of the paper. This is always true, of course, but particularly so for IDF papers. Theoreticians gravitate to IDF models precisely because of their tractability and simplicity of assumptions. Even slight departures from the most simple underlying assumptions can dramatically complicate model prediction, as my lab knows from direct experience (Avgar et al. 2021). It is unworthy to

complain that nature doesn't comply with our theoretical preference for simplicity. Provision of empirical examples, even those with potentially serious flaws, help readers recognize ways that important concepts might be useful in better understanding their own study systems and encourage useful follow-up studies. They should rarely be treated as logical "proofs" per se.

References:

Avgar, T., G.S. Betini, and J.M. Fryxell. 2020. Habitat selection patterns are density dependent under the ideal free distribution. *Journal of Animal Ecology* DOI: 10.1111/1365-2656.13352
DiNuzzo ER, Griffen BD. 2020 The effects of animal personality on the ideal free distribution. *Proc. R. Soc. B* 287:20201095.

Review form: Reviewer 3

Recommendation

Major revision is needed (please make suggestions in comments)

Scientific importance: Is the manuscript an original and important contribution to its field?

Good

General interest: Is the paper of sufficient general interest?

Excellent

Quality of the paper: Is the overall quality of the paper suitable?

Good

Is the length of the paper justified?

Yes

Should the paper be seen by a specialist statistical reviewer?

No

Do you have any concerns about statistical analyses in this paper? If so, please specify them explicitly in your report.

No

It is a condition of publication that authors make their supporting data, code and materials available - either as supplementary material or hosted in an external repository. Please rate, if applicable, the supporting data on the following criteria.

Is it accessible?

N/A

Is it clear?

N/A

Is it adequate?

N/A

Do you have any ethical concerns with this paper?

No

Comments to the Author

In this manuscript, the authors comment on a recently published article by DiNuzzo & Griffen (hereafter D&G, for concision) in Proceedings B entitled "The effects of animal personality on the ideal free distribution". I am pleased to see this comment come through. It is timely and, I think, an important contribution to this discussion. Because of the brevity of the comment, I will address each of the authors' main 5 points (which, I think, should be better identified instead of using simple parenthetical numbers).

1. Interpretation of the results in terms of personality variation

I think the authors are spot-on in their first point. Because all animals in the D&G simulation have perfect knowledge of the landscape, and because some animals move more slowly, the IFD is achieved but slowly. This is, as they say, due to inefficient movement, and this point is reflected across several of my reviews of D&G's original work. If we accept that variation in movement rates constitutes "personality" (a common assertion among wildlife ecologists, for example, for better or worse), then this is not necessarily the problem the authors assert. Still, their observations are correct, leading D&G's model to produce what I have referred to as a "self-fulfilling prophecy". This seems obvious to me, but many readers may not pick up on this, and so here the comment is providing an important observation about D&G's work.

However, I think their second paragraph in this section is in some ways seeking a fight where one is not necessary. D&G's work was not intended to discuss the eco-evolutionary causes of personality and the IFD, but rather to address undermatching and the role of variation in personality (there expressed as variation in movement rates) in shaping the rate at which an IFD is achieved. Nothing the authors say in that paragraph is technically wrong, but their points could not be adequately addressed in the D&G model because stable coexistence of different personalities would require a dedicated game-theoretic-styled model wherein different personalities (there, movement strategies) could be evaluated for their individual fitness. I agree with the authors' implication here that this would be a valuable contribution, but criticizing the D&G study for omitting something like this is inappropriate as it was outside the scope of the original work.

2. Deficiencies in the technical implementation of the model

I really appreciate the reference to a Zenodo page laying out all code and highlighting errors in the code and thank the authors for providing this.

3. The effects of population size on deviations from the IFD

This is a good catch, and an elegant way to demonstrate the problems caused by treating the population size problem in the manner of D&G given how their algorithm works. I suspect that one could also approximately re-create the authors' Figure 1 using the original outputs of the D&G work by simply dividing the time to achieve the IFD by the number of individuals in the population; if so, that would be an even more powerful demonstration.

Additionally, their note that when individuals occupy patches simultaneously they benefit by staying if others leave is well-established. There is a fair amount of literature demonstrating this implicitly by virtue of the mathematics at play (the various works of Cantrell, Cressman, and Cosner come to mind).

4. The measure used for quantifying deviations from the IFD

Again, there are some good observations here that are driven by the simplicity of the original D&G model. The authors are correct that time-to-IFD is not a flawless metric given how individual discrepancies from an optimal movement strategy and initial distributions of the population can affect emergence of the IFD. However, their third point that time-to-IFD is only

sensible when an IFD can be achieved seems somewhat unnecessary to me, since that is the condition imposed by the original D&G model. In this third point, the authors are in fact arguing not for a different metric of IFD (e.g. variation in intake rates), but rather for a fully different parameterization of the model and its rules. I would suggest this point be revised to make this more explicit because, frankly, I think the authors are wrong to suggest that variation in intake rates is superior to time-to-IFD in this instance. Yes, at IFD intakes rates will be identical across patches and individuals, but the spatial consequence of that is more animals in high quality patches, which is how D&G represented their original work (as I recall).

Still, if you change the way the model works by including e.g. seasonality in patch quality, then yes, a different metric as suggested by the authors could be much more appropriate. Hence my suggestion that this section be revised to reflect model structure more explicitly, and particularly how model structure affects what metrics are most appropriate.

5. The relevance of the empirical data set

Excellent points. My only suggestion here is that the authors might make reference to other empirical studies highlighting the difficulty in evaluating IFD in real animals due to the various problems identified by the authors here.

Minor Comments:

Please add line numbers so that we can pointedly address specific portions of the manuscript more efficiently.

Decision letter (RSPB-2021-0367.R0)

21-Mar-2021

Dear Mr Netz:

I am writing to inform you that your manuscript RSPB-2021-0367 entitled "Details matter when modelling the effects of animal personality on the spatial distribution of foragers" has, in its current form, been rejected for publication in Proceedings B. All of the reviewers appreciated the goals of your paper, but had a number of suggestions for necessary edits. They do an excellent job of explaining them, so I will not repeat them here. We would be happy to consider a resubmitted manuscript if you are able to fully address each of their concerns. Please note, however, that this is not a provisional acceptance.

- 1) A 'response to referees' document including details of how you have responded to the comments, and the adjustments you have made.
- 2) A clean copy of the manuscript and one with 'tracked changes' indicating your 'response to referees' comments document.

- 3) Line numbers in your main document.
 4) Data - please see our policies on data sharing to ensure that you are complying (<https://royalsociety.org/journals/authors/author-guidelines/#data>).

Sincerely,
 Dr Sarah Brosnan
 Editor, Proceedings B
 mailto: proceedingsb@royalsociety.org

Associate Editor
 Board Member: 1
 Comments to Author:

The authors note some limits to the assumptions built into the model of the target article and point out additional conditionalities to results interpretation depending on ecological context. All reviewers found valuable points in the commentary, but also several areas requiring substantial improvement. Close and thorough attention to the reviewers' excellent comments is needed for further consideration.

Reviewer(s)' Comments to Author:
 Referee: 1
 Comments to the Author(s)

In the article, "Details matter when modelling the effects of animal personality on the spatial distribution of foragers" the authors provide a critique of a recent publication and model produced in DiNuzzo and Griffen (2020). The authors provide some good comments on how assumptions were built into the model, highlight some errors, and touch upon how results interpretation may change depending on the ecological context. However, there are a number of issues with this critique and points that seem to have been misunderstood in the original paper. For instance, the authors seem to not understand why a Type II functional response limiting max consumption rate was used and why it caused results patterns to reverse. Additionally, the authors state a number of times that changes to the model may alter findings, but frequently do not actually demonstrate how the findings would be altered. This manuscript should focus more on specifically describing and expanding upon how model differences affect ecological interpretation, and demonstrate with simulations how results would differ when coding/parameters are changed.

Specific comments below

Point 1

The authors make a good point that DiNuzzo and Griffen's model shows that the presence of inefficient movers is the underlying cause behind populations taking longer to reach the IFD in this model. However, the authors seem to be assuming that DiNuzzo and Griffen are advocating that variation in personality almost always slows down the approach to IFD when the paper is more a demonstration of how personalities CAN slow down the approach to IFD and provide a simple example. There are a number of reasons why individuals may not be equal in movement efficiency (e.g. injury, sickness, physiological differences). Similarly, there are a number of ways that individual personality can affect movement. The most apparent personality effect is that shy, docile individuals often spend more time in refuge and less time active, even when hungry in many circumstances. However, in a number of species, shy individuals often have smaller home ranges (e.g. Watson & Miller 1971; Spiegel et al. 2015), shorter dispersal distances (e.g. Cote et al.

2010), or slower reaction time to changing food availability (e.g. van Overveld and Matthysen 2010) than bold individuals which can preclude them from finding optimal patches. Differences in competitive ability are already well-known to be a cause for deviations from IFD, but differences in personality are also often the reason why individuals are not equally competitive. Although not discussed by DiNuzzo and Griffen, boldness has the potential to do the reverse where exceptionally fast individuals may quickly find resources and help the population reach IFD sooner, particularly if they attract conspecifics or communicate (e.g. bees).

Thus, DiNuzzo and Griffen were not so much flawed in the interpretation of their model findings as the interpretation could be extended to include anything that causes individuals to be inefficient in their movement. Should reduce the criticism here, but continue to point out how it is the presence of inefficient movers and factors that govern movement that retard approach to IFD.

Cote, J., Clobert, J., Brodin, T., Fogarty, S. & Sih, A. (2010). Personality-dependent dispersal: characterization, ontogeny and consequences for spatially structured populations. *Philos. Trans. R. Soc. Lond. B. Biol. Sci.*, 365, 4065–76

Spiegel, O., Leu, S.T., Sih, A., Godfrey, S.S. & Bull, C.M. (2015). When the going gets tough: behavioural type-dependent space use in the sleepy lizard changes as the season dries. *Proc. R. Soc. B Biol. Sci.*, 282, 20151768

van Overveld, T. & Matthysen, E. (2010). Personality predicts spatial responses to food manipulations in free-ranging great tits (*Parus major*). *Biol. Lett.*, 6, 187–90

Watson, B.Y.A. & Miller, G.R. (1971). Territory Size and Aggression in a Fluctuating Red Grouse Population. *J. Anim. Ecol.*, 40, 367–383

First point of the third paragraph: The authors question why different activity levels can stably coexist within a system, but that seems to be beyond the scope of the study for DiNuzzo and Griffen which was to show how personality variation can be responsible for IFD undermatching. There is an entire section of literature on why personalities may persist within a population. The quick answer is that all models are simplifications of complex systems and there are outside factors not considered here such as predation rate of different individuals (shy individuals have higher survival probability than bold, but bold can reproduce more because they gain more resources). Sih et al. (2015) provides a good overview on a number of potential mechanisms that may lead to populations maintaining multiple personality types across either short or long time scales (lifetime to multigenerational).

The second point of this third paragraph is a good one and ties into the first point. The authors should expand upon this idea that equilibrium states vary depending on diversity of individuals within a population and other external factors beyond resource availability. Providing specific examples, scenarios, and/or illustrations and models quantifying how much these equilibrium states should change depending on these factors would be valuable here.

Sih, A., Mathot, K. J., Moirón, M., Montiglio, P. O., Wolf, M., & Dingemanse, N. J. (2015). Animal personality and state-behaviour feedbacks: a review and guide for empiricists. *Trends in Ecology & Evolution*, 30(1), 50-60.

Point 2

In this section the authors outline some of their issues with the coding of the model and refer to a citation written by them and posted to github providing specific details on their model issues (citation 8). The technical notes of this citation need to be provided as supplementary material for this article and a url to the overall work should also be provided in the citation.

In these technical notes, the authors note some important points in the code of DiNuzzo and Griffen, but also themselves seem to have had some misunderstandings which are outlined below.

Note 2: “set quality (2 + random 8)” would have quality values go from 2 – 10 with 9 distinct levels of habitat quality, not 2 – 9 as the authors state.

Note 3: The authors make a great point here that designating 50 time steps as a stopping point may cause the simulation to end prematurely before IFD is reached and that this may be the reason for the hump in depicted in Figure 4e of DiNuzzo and Griffen. The authors state in their manuscript that in their model they no longer see this hump, but they do not provide a figure of this since Figure 1 of their paper does not ever adjust the personality types to be 0.9 and 0.1 rather 0.8 and 0.2.

Note 6: Here, the authors are concerned that there is an upper limit to the max feeding rate, set at 1, and that this causes a “bug” in the system where individuals choose not to move to patches with higher number of resources because they are on a patch that maxes their resource consumption limit, even though it has fewer overall resources than the richest available patch. This constitutes a major point of the authors’ submitted article, but the authors do not seem to have understood the reason behind also running the model using a Type II functional response in addition to simulations with a Type I functional response. Type I functional responses are rare in nature because they assume that individuals are never satiated and will consume whatever resources are available. However, eventually individuals’ needs are met and they stop consuming resources even when more are available in the region. Under these scenarios, there is no point for individuals to expend energy traveling to another location with more resources, potentially risking higher predation rates during travel, if they can already acquire all the resources they need in the present location. It is therefore more realistic to examine these scenarios using a type II functional response where individual feeding rate is capped and where individuals stop moving on patches that meet their needs but are not the richest in available resources. However, the increase in time to reach IFD with more active individuals is not actually so much a result of individuals taking more time to find the richest available patch because they are satiated on “suboptimal patches”, but because of changes to how patch resources are divided among individuals. Resources on a patch are no longer divided into ever smaller infinitesimal fractions as crabs are added (where a patch with 8 resources could theoretically support 4 crabs each consuming 2 resources or 400 crabs each consuming 0.02 resources, which is unrealistic), but instead resources are subtracted as crabs are added so eventually no more crabs can go on a certain patch because all resources would already be consumed (a patch with 8 resources could support 8 crabs consuming 1 resource; code lines 80 and 98 versus 89 and 107). This means that adding a crab to a patch has a much larger effect on available resources and can cause a chain reaction in additional movement as a single crab is added to a patch that had few available resources remaining. As the number of active individuals increase, so too would the probability that a chain reaction in movement would occur.

This difference in how resources are divided among population members and its effect on findings should have been highlighted better by DiNuzzo and Griffen since both calculations are valid under different ecological circumstances and produce very different results. But framing the results of Figure S1 as an error or artefact does not seem appropriate. It could however lend itself well to a discussion on ecological context and methods of modeling that context. Especially, if the authors run a sensitivity analyses varying the max feeding limit and report subsequent results. If the authors want to frame this as an error then they need to show that the results become qualitatively different when the mistakes highlighted in points 4 and 5 are corrected and should provide new model code in supplemental.

In the supplemental material authors also need to provide the code used to produce the model results depicted in Figure 1 as well as highlight the associated changes they made to the original model code of DiNuzzo and Griffen.

Point 3

Again, code for Figure 1 would best be provided in supplemental. Note that DiNuzzo and Griffen used 1000 replicate simulations rather than the 300 used here.

Point 4

The authors are correct that time to reach IFD is an imperfect measure for quantifying such deviations in a natural system. This method is really only suitable for a highly standardized model like the one utilized by DiNuzzo and Griffen that only implements 2 personality types and runs 1000 simulations/scenario to help identify the variation in results due to differences in initial starting conditions and probability that simulation random-float numbers hit movement thresholds. The authors should focus this segment more on alternative methods to measure differences from IFD that work better for natural populations or more sophisticated models. This section would benefit greatly if the authors implement these different measurements of deviation from IFD in their own models and compare the results. Does using variance in intake rates across patches actually lead to different conclusions?

Point 5

It is true that refuge use does not necessarily translate to activity level as individuals can still be moving around to some degree while within a refuge. However, the authors seem to have misinterpreted how DiNuzzo and Griffen relate refuge use (what they call activity level) to their calculation of movement propensity. Here, refuge use was not treated as a 1-1 relationship to movement propensity but instead relied upon a prior study conducted by Belgrad and Griffen (2018) that quantified the relationship between individual refuge use and movement propensity to determine how much crabs would move in the model. The major criticism here then should not be that DiNuzzo and Griffen used the wrong study system since there is a fair amount of literature demonstrating that the species exhibits persistent individual differences in various behaviors that correlate with a number of other important characteristics. Rather the major point of concern is the number of relationships estimated in assigning population movement propensities for the model and the extra level of uncertainty built into their calculations. Instead of using movement propensities measured from a large sample of the population, DiNuzzo and Griffen estimate individual refuge use propensity from a size distribution of the population and then use that estimation to estimate movement propensity of the population. The authors should instead focus their critique on this level of uncertainty or that DiNuzzo and Griffen needed to use a study system where personality distribution did not have to be estimated.

Figures 3 and 4 provide graphs of how the degree of personality variation can affect the time to reach IFD from theoretical populations. The purpose of Figure 5, does not appear to really be about illustrating how variation in personality can affect the time to IFD which they do with their theoretical populations. Figure 5 is a rough calculation of how much longer it should take a natural population to reach IFD given that individuals do not follow “ideal” movement probabilities. If one were to vary the population personality distributions for such a calculation, it would no longer be a real-world personality distribution. Is it natural for 90% of individuals to have an activity level of 0.9? It would have been better if the DiNuzzo and Griffen paper used multiple populations to provide that information, but that also seems to be the point of the first part of their manuscript. Studies almost never report the personality distribution of populations and there is a lack of data on this subject. Still, they did find 10 studies that do seem to report that information so they could have potentially used that data to improve their estimates further.

Belgrad, B. A., & Griffen, B. D. (2018). Personality interacts with habitat quality to govern individual mortality and dispersal patterns. *Ecol. Evol.*, 8, 7216-7227.

Referee: 2

Comments to the Author(s)

This manuscript clarifies and enriches some potentially key processes first outlined by DiNuzzo and Griffen (2020) influencing the time required for hypothetical populations to converge on an

ideal free distribution and the commonly-observed tendency for under-matching to occur as a result. The authors central thesis is that any delays in decision-making will automatically extend the time to equilibrium by definition. This seems a subtle, but important, point since other many factors (e.g. variation among individuals due to differences due to internal motivation, physiological state, perception, or cognition) could act in this way in addition to variation in personality syndrome.

The authors also clarify that a key component in any such model is whether decisions are made sequentially by individuals across the population or simultaneously. Changing this assumption changed several outcomes in important ways. Such considerations about discrete vs continuous representations of time can fundamentally alter outcomes in many models in ecology, including the most basic of density-dependent formulations. Given that we can rarely (perhaps never) be sure that either continuous or discrete formulations are truly correct, it is important to recognize this as a key biological constraint on all model formulations. The authors provision of code for IDF formulation according to simultaneous decision-making is a most useful contribution to the field.

I am less enthusiastic about the authors' complaint that DiNuzzo and Griffen's (2020) crab population represents a less than perfect example of the processes discussed in the theoretical part of the paper. This is always true, of course, but particularly so for IDF papers. Theoreticians gravitate to IDF models precisely because of their tractability and simplicity of assumptions. Even slight departures from the most simple underlying assumptions can dramatically complicate model prediction, as my lab knows from direct experience (Avgar et al. 2021). It is unworthy to complain that nature doesn't comply with our theoretical preference for simplicity. Provision of empirical examples, even those with potentially serious flaws, help readers recognize ways that important concepts might be useful in better understanding their own study systems and encourage useful follow-up studies. They should rarely be treated as logical "proofs" per se.

References:

Avgar, T., G.S. Betini, and J.M. Fryxell. 2020. Habitat selection patterns are density dependent under the ideal free distribution. *Journal of Animal Ecology* DOI: 10.1111/1365-2656.13352

DiNuzzo ER, Griffen BD. 2020 The effects of animal personality on the ideal free distribution. *Proc. R. Soc. B* 287:20201095.

Referee: 3

Comments to the Author(s)

In this manuscript, the authors comment on a recently published article by DiNuzzo & Griffen (hereafter D&G, for concision) in *Proceedings B* entitled "The effects of animal personality on the ideal free distribution". I am pleased to see this comment come through. It is timely and, I think, an important contribution to this discussion. Because of the brevity of the comment, I will address each of the authors' main 5 points (which, I think, should be better identified instead of using simple parenthetical numbers).

1. Interpretation of the results in terms of personality variation

I think the authors are spot-on in their first point. Because all animals in the D&G simulation have perfect knowledge of the landscape, and because some animals move more slowly, the IFD is achieved but slowly. This is, as they say, due to inefficient movement, and this point is reflected across several of my reviews of D&G's original work. If we accept that variation in movement rates constitutes "personality" (a common assertion among wildlife ecologists, for example, for better or worse), then this is not necessarily the problem the authors assert. Still, their observations are correct, leading D&G's model to produce what I have referred to as a "self-

fulfilling prophecy". This seems obvious to me, but many readers may not pick up on this, and so here the comment is providing an important observation about D&G's work.

However, I think their second paragraph in this section is in some ways seeking a fight where one is not necessary. D&G's work was not intended to discuss the eco-evolutionary causes of personality and the IFD, but rather to address undermatching and the role of variation in personality (there expressed as variation in movement rates) in shaping the rate at which an IFD is achieved. Nothing the authors say in that paragraph is technically wrong, but their points could not be adequately addressed in the D&G model because stable coexistence of different personalities would require a dedicated game-theoretic-styled model wherein different personalities (there, movement strategies) could be evaluated for their individual fitness. I agree with the authors implication here that this would be a valuable contribution, but criticizing the D&G study for omitting something like this is inappropriate as it was outside the scope of the original work.

2. Deficiencies in the technical implementation of the model

I really appreciate the reference to a Zenodo page laying out all code and highlighting errors in the code and thank the authors for providing this.

3. The effects of population size on deviations from the IFD

This is a good catch, and an elegant way to demonstrate the problems caused by treating the population size problem in the manner of D&G given how their algorithm works. I suspect that one could also approximately re-create the authors' Figure 1 using the original outputs of the D&G work by simply dividing the time to achieve the IFD by the number of individuals in the population; if so, that would be an even more powerful demonstration.

Additionally, their note that when individuals occupy patches simultaneously they benefit by staying if others leave is well-established. There is a fair amount of literature demonstrating this implicitly by virtue of the mathematics at play (the various works of Cantrell, Cressman, and Cosner come to mind).

4. The measure used for quantifying deviations from the IFD

Again, there are some good observations here that are driven by the simplicity of the original D&G model. The authors are correct that time-to-IFD is not a flawless metric given how individual discrepancies from an optimal movement strategy and initial distributions of the population can affect emergence of the IFD. However, their third point that time-to-IFD is only sensible when an IFD can be achieved seems somewhat unnecessary to me, since that is the condition imposed by the original D&G model. In this third point, the authors are in fact arguing not for a different metric of IFD (e.g. variation in intake rates), but rather for a fully different parameterization of the model and its rules. I would suggest this point be revised to make this more explicit because, frankly, I think the authors are wrong to suggest that variation in intake rates is superior to time-to-IFD in this instance. Yes, at IFD intakes rates will be identical across patches and individuals, but the spatial consequence of that is more animals in high quality patches, which is how D&G represented their original work (as I recall).

Still, if you change the way the model works by including e.g. seasonality in patch quality, then yes, a different metric as suggested by the authors could be much more appropriate. Hence my suggestion that this section be revised to reflect model structure more explicitly, and particularly how model structure affects what metrics are most appropriate.

5. The relevance of the empirical data set

Excellent points. My only suggestion here is that the authors might make reference to other empirical studies highlighting the difficulty in evaluating IFD in real animals due to the various problems identified by the authors here.

Minor Comments:

Please add line numbers so that we can pointedly address specific portions of the manuscript more efficiently.

Author's Response to Decision Letter for (RSPB-2021-0367.R0)

See Appendix A.

RSPB-2021-0903.R0

Review form: Reviewer 1

Recommendation

Accept with minor revision (please list in comments)

Scientific importance: Is the manuscript an original and important contribution to its field?

Acceptable

General interest: Is the paper of sufficient general interest?

Marginal

Quality of the paper: Is the overall quality of the paper suitable?

Good

Is the length of the paper justified?

Yes

Should the paper be seen by a specialist statistical reviewer?

No

Do you have any concerns about statistical analyses in this paper? If so, please specify them explicitly in your report.

No

It is a condition of publication that authors make their supporting data, code and materials available - either as supplementary material or hosted in an external repository. Please rate, if applicable, the supporting data on the following criteria.

Is it accessible?

Yes

Is it clear?

Yes

Is it adequate?

Yes

Do you have any ethical concerns with this paper?

No

Comments to the Author

Netz et al. present a revised paper on “Details matter when modelling the effects of animal personality on the spatial distribution of foragers”. This draft is substantially improved upon, especially when illustrating the errors in the code of DiNuzzo and Griffen (D+G) and its consequences. I only have two concerns that are relatively easy to remedy over Netz et al.’s critiques of the paper which both seem to be a misunderstanding of the focus of D+G’s paper and figures. Described below.

Lines 24-28: This critique about D+G paper seems to be missing the point about the original manuscript and should still be reduced. The D+G paper is not focusing so much on variation in personality types as Netz et al. describe, but on the existence of personalities itself (where individuals are not completely flexible behaviorally in being able to respond to their environment). The D+G paper is demonstrating that the presence of a personality, particularly an inactive personality, “may inhibit the ability of a population to track changes in habitat quality”. The point is that a personality could change the time to reach IFD at all and that it is important to know the variation in personalities within a population because this will help determine the time it takes to reach IFD. The D+G paper is not at all about personality variation itself increasing time to IFD as Netz et al. seem to focus upon. However, a critique could be added that D+G focus on how that inflexible behavior slows time to matching and that there may be mechanisms where organisms can communicate or observe each other to speed the approach to IDF although that would involve a completely different spectrum of personality. If one wants to argue this, then the authors should elaborate their critique to explain how D+G focus only on activity level as their representative of personality and that other behavioral facets such as sociability and numbers of leaders or communicators may speed the approach to IDF (with examples of systems where this may occur).

Lines 149-163: Again Netz et al. are focusing on the idea that D+G are using Figure 5 to demonstrate the role of personality variation on time to reach the IFD. This is not the case since they already illustrate how personality variation can affect time to IFD with their earlier simulations of artificial personality distributions (e.g. D+G Figure 3). Figure 5 is used to show how much longer it should take for a natural population with a “known” personality distribution (calculated from size) to reach IFD over a theoretical population that does not have personalities and where each individual is perfectly flexible in their behavior and active. The calculation takes the observed variation into account by looking at the variation in body size within the population, transforming that into personality from previously made regressions of body size and personality (so x individuals have personality 0.01, y individuals have 0.014, z individuals have 0.2... etc.) then running the simulation to see how long that population takes to reach IFD with that natural personality distribution. It’s a real-world example of how much time inflexible behavior in a natural population may add to reaching IDF and a demonstration of how their model or a model built for a specific study system could be applied to other populations. It would have been nice if D+G did this with more populations as comparison or looked at subpopulations, but was not necessary to prove their point.

Decision letter (RSPB-2021-0903.R0)

14-May-2021

Dear Mr Netz:

Your manuscript has now been peer reviewed and the review has been assessed by an Associate Editor. The reviewer, AE, and I appreciate the extensive revisions that you have undertaken, however the reviewer highlights two more areas in which the comment mis-represents the original paper's goal. Please revise your manuscript to take into account these two issues. The reviewer's comments (not including confidential comments to the Editor) and the comments from the Associate Editor are included at the end of this email for your reference.

Research ethics:

Use of animals and field studies:

It is a condition of publication that you make available the data and research materials supporting the results in the article (<https://royalsociety.org/journals/authors/author-guidelines/#data>). Datasets should be deposited in an appropriate publicly available repository and details of the associated accession number, link or DOI to the datasets must be included in the Data Accessibility section of the article (<https://royalsociety.org/journals/ethics-policies/data-sharing-mining/>). Reference(s) to datasets should also be included in the reference list of the article with DOIs (where available).

If you wish to submit your data to Dryad (<http://datadryad.org/>) and have not already done so you can submit your data via this link [http://datadryad.org/submit?journalID=RSPB&manu=\(Document not available\)](http://datadryad.org/submit?journalID=RSPB&manu=(Document%20not%20available)), which will take you to your unique entry in the Dryad repository.

Please submit a copy of your revised paper within three weeks. If we do not hear from you within this time your manuscript will be rejected. If you are unable to meet this deadline please let us know as soon as possible, as we may be able to grant a short extension.

Best wishes,
Dr Sarah Brosnan
Editor, Proceedings B
mailto: proceedingsb@royalsociety.org

Associate Editor

Comments to Author:

The authors have made extensive changes in response to the original reviewers' comments and the contribution is much improved. There remain a few areas that require further modification (see valuable insights provided in the review), especially regarding the need to be precise when discussing and evaluating the claims in the target article.

Reviewer(s)' Comments to Author:

Referee: 1

Comments to the Author(s).

Netz et al. present a revised paper on "Details matter when modelling the effects of animal personality on the spatial distribution of foragers". This draft is substantially improved upon, especially when illustrating the errors in the code of DiNuzzo and Griffen (D+G) and its consequences. I only have two concerns that are relatively easy to remedy over Netz et al.'s critiques of the paper which both seem to be a misunderstanding of the focus of D+G's paper and figures. Described below.

Lines 24-28: This critique about D+G paper seems to be missing the point about the original manuscript and should still be reduced. The D+G paper is not focusing so much on variation in personality types as Netz et al. describe, but on the existence of personalities itself (where

individuals are not completely flexible behaviorally in being able to respond to their environment). The D+G paper is demonstrating that the presence of a personality, particularly an inactive personality, “may inhibit the ability of a population to track changes in habitat quality”. The point is that a personality could change the time to reach IFD at all and that it is important to know the variation in personalities within a population because this will help determine the time it takes to reach IFD. The D+G paper is not at all about personality variation itself increasing time to IFD as Netz et al. seem to focus upon. However, a critique could be added that D+G focus on how that inflexible behavior slows time to matching and that there may be mechanisms where organisms can communicate or observe each other to speed the approach to IDF although that would involve a completely different spectrum of personality. If one wants to argue this, then the authors should elaborate their critique to explain how D+G focus only on activity level as their representative of personality and that other behavioral facets such as sociability and numbers of leaders or communicators may speed the approach to IDF (with examples of systems where this may occur).

Lines 149-163: Again Netz et al. are focusing on the idea that D+G are using Figure 5 to demonstrate the role of personality variation on time to reach the IFD. This is not the case since they already illustrate how personality variation can affect time to IFD with their earlier simulations of artificial personality distributions (e.g. D+G Figure 3). Figure 5 is used to show how much longer it should take for a natural population with a “known” personality distribution (calculated from size) to reach IFD over a theoretical population that does not have personalities and where each individual is perfectly flexible in their behavior and active. The calculation takes the observed variation into account by looking at the variation in body size within the population, transforming that into personality from previously made regressions of body size and personality (so x individuals have personality 0.01, y individuals have 0.014, z individuals have 0.2.... etc.) then running the simulation to see how long that population takes to reach IFD with that natural personality distribution. It’s a real-world example of how much time inflexible behavior in a natural population may add to reaching IDF and a demonstration of how their model or a model built for a specific study system could be applied to other populations. It would have been nice if D+G did this with more populations as comparison or looked at subpopulations, but was not necessary to prove their point.

Author's Response to Decision Letter for (RSPB-2021-0903.R0)

See Appendix B.

Decision letter (RSPB-2021-0903.R1)

01-Jun-2021

Dear Mr Netz

I am pleased to inform you that your manuscript entitled "Details matter when modelling the effects of animal personality on the spatial distribution of foragers" has been accepted for publication in Proceedings B.

Data Accessibility section

Open Access

Your article has been estimated as being 3.5 pages long. Our Production Office will be able to confirm the exact length at proof stage.

Paper charges

Sincerely,

Dr Sarah Brosnan

Associate Editor:

Comments to Author:

Thank you for your attention to the reviewer comments and attendant modifications to the commentary. These additional changes have improved the clarity of the work.

Appendix A

Details matter when modelling the effects of animal personality on the spatial distribution of foragers

Christoph Netz, Aparajitha Ramesh, Jakob Gismann, Pratik R. Gupte, Franz J. Weissing

Response to Referees

Overview of the most important changes

We thank all three referees and the associate editor for their comments on the first version of our submission. Their feedback was very useful and has resulted in a revised version that, in our opinion, makes its points in a more transparent manner. Below, we will respond in detail to the referees' comments. Here, we will give a brief summary of the most significant changes:

- In response to Referee 1, we have added two figures that demonstrate how key results in the article of DiNuzzo and Griffen (2020) change, both quantitatively and qualitatively, when their erroneous NetLogo code is replaced by a corrected version. We added the corrected code to the supplement.
- In response to Referee 2, we have clarified that our critique of the empirical example discussed by DiNuzzo and Griffen is not aimed at the deviation between empirical data and theoretically formulated processes (which is natural), but rather at the analysis and processing of the empirical data.
- In response to Referee 3, we have moved all evolutionary considerations from the core of the manuscript to a brief outlook section at the end.

Response to the Associate Editor:

The authors note some limits to the assumptions built into the model of the target article and point out additional conditionalities to results interpretation depending on ecological context. All reviewers found valuable points in the commentary, but also several areas requiring substantial improvement. Close and thorough attention to the reviewers' excellent comments is needed for further consideration.

As indicated above, we have substantially improved the manuscript in response to the reviewers' comments.

Response to Referee 1:

In the article, "Details matter when modelling the effects of animal personality on the spatial distribution of foragers" the authors provide a critique of a recent publication and model produced in DiNuzzo and Griffen (2020). The authors provide some good comments on how assumptions were built into the model, highlight some errors, and touch upon how results interpretation may change depending on the ecological context. However, there are a number of issues with this critique and points that seem to have been misunderstood in the original paper. For instance, the authors seem to not understand why a Type II functional response limiting max consumption rate was used and why it caused results patterns to reverse. Additionally, the authors state a number of times that changes to the model may alter findings, but frequently do not actually demonstrate how the findings would be altered. This manuscript should focus more on specifically describing and expanding upon how model differences affect ecological interpretation, and demonstrate with simulations how results would differ when coding/parameters are changed.

First of all, we would like to thank Referee 1 for the time and effort invested into their comprehensive report. More specifically, we are thankful for the advice to demonstrate more clearly in which way, and to what extent, the findings of DiNuzzo and Griffen are perturbed by errors in their code. To this end, we have now included two new figures (Figs 1 and 2) that contrast the simulation outcomes in DiNuzzo and Griffen (2020) (based on their erroneous NetLogo code) with the outcome of simulations that make use of a corrected version of the code. We also demonstrate how mistakes in the code induced the simulations of DiNuzzo and Griffen to stop prematurely in a biased manner, and sometimes in more than 95% of the cases.

We fully agree with Referee 1 that a type II functional response is a much more realistic modelling assumption than the unlimited linear functional response used by DiNuzzo and Griffen in their main text. This, however, is not the reason why DiNuzzo and Griffen found a different pattern in their Figure S1 (where they consider a type II functional response) than in their main text. In fact, we show mathematically that the two functional response curves used by DiNuzzo and Griffen have the property that they lead to the same ranking of patches and, hence, that they should lead to identical simulation outcomes. This is independent proof that Figure S1 does not reflect the more realistic assumption of a type II functional response curve, but a serious mistake in the NetLogo code. Indeed, the new Figure 1B shows that, after correcting the code, simulations based on a type II functional response show the same of pattern as simulations based on an unlimited linear functional response.

In the previous version, we pointed out the mistake in DiNuzzo and Griffen's NetLogo code in considerable detail, but this explanation was "hidden away" in our Technical Discussion document while the description in our main text was a bit vague. Thanks to the comments of Referee 1 we became aware of this and clarified our treatment on this point.

Point 1

The authors make a good point that DiNuzzo and Griffen's model shows that the presence of inefficient movers is the underlying cause behind populations taking longer to reach the IFD in this model. However, the authors seem to be assuming that DiNuzzo and Griffen are advocating that variation in personality almost always slows down the approach to IFD when the paper is more a demonstration of how personalities CAN slow down the approach to IFD and provide a simple example. There are a number of reasons why individuals may not be equal in movement efficiency (e.g. injury, sickness, physiological differences). Similarly, there are a number of ways that individual personality can affect movement. The most apparent personality effect is that shy, docile individuals often spend more time in refuge and less time active, even when hungry in many circumstances. However, in a number of species, shy individuals often have smaller home ranges (e.g. Watson & Miller 1971; Spiegel et al. 2015), shorter dispersal distances (e.g. Cote et al. 2010), or slower reaction time to changing food availability (e.g. van Overveld and Matthysen 2010) than bold individuals which can preclude them from finding optimal patches. Differences in competitive ability are already well-known to be a cause for deviations from IFD, but differences in personality are also often the reason why individuals are not equally competitive. Although not discussed by DiNuzzo and Griffen, boldness has the potential to do the reverse where exceptionally fast individuals may quickly find resources and help the population reach IFD sooner, particularly if they attract conspecifics or communicate (e.g. bees).

Thus, DiNuzzo and Griffen were not so much flawed in the interpretation of their model findings as the interpretation could be extended to include anything that causes individuals to be inefficient in their movement. Should reduce the criticism here, but continue to point out how it is the presence of inefficient movers and factors that govern movement that retard approach to IFD.

We wholeheartedly agree with all these comments. Our main aim in our first point was to clarify that low activity levels generally will retard the redistribution towards the IFD, and that primarily not personality differences are responsible for the findings of DiNuzzo and Griffen but rather differences in the overall activity level. Our last sentence in this paragraph is then intended to ward off the misunderstanding that personality variation *per se* slows down the approach to the IFD, the same misunderstanding that Referee 1 so convincingly argues against. We feel justified in warning against this, as on various occasions DiNuzzo and Griffen seem to extrapolate their findings beyond the context of variation in “activity” levels. This is exemplified by the title and the next-to-final sentence in their abstract: *“We suggest that animal personality, particularly the prevalence of inactive personality types, may inhibit the ability of a population to track changes in habitat quality, therefore leading to undermatching of the IFD.”*

First point of the third paragraph: The authors question why different activity levels can stably coexist within a system, but that seems to be beyond the scope of the study for DiNuzzo and Griffen which was to show how personality variation can be responsible for IFD undermatching. There is an entire section of literature on why personalities may persist within a population. The quick answer is that all models are simplifications of complex systems and there are outside factors not considered here such as predation rate of different individuals (shy individuals have higher survival probability than bold, but bold can reproduce more because they gain more resources). Sih et al. (2015) provides a good overview on a number of potential mechanisms that may lead to populations maintaining multiple personality types across either short or long time scales (lifetime to multigenerational).

The second point of this third paragraph is a good one and ties into the first point. The authors should expand upon this idea that equilibrium states vary depending on diversity of individuals within a population and other external factors beyond resource availability. Providing specific examples, scenarios, and/or illustrations and models quantifying how much these equilibrium states should change depending on these factors would be valuable here.

We agree with the first point (which coincides with the first concern of Referee 3). Accordingly, we have removed our evolutionary considerations from this paragraph and shifted them to a more general “outlook” section at the end of our manuscript. We preserved our second point but refrained from further expanding on this here, as this will be the subject of a future study.

Point 2

In this section the authors outline some of their issues with the coding of the model and refer to a citation written by them and posted to github providing specific details on their model issues (citation 8). The technical notes of this citation need to be provided as supplementary material for this article and a url to the overall work should also be provided in the citation.

We have shifted the technical notes to the supplementary material and now also provide a link to the online repository.

In these technical notes, the authors note some important points in the code of DiNuzzo and Griffen, but also themselves seem to have had some misunderstandings which are outlined below. Note 2: “set quality (2 + random 8)” would have quality values go from 2 – 10 with 9 distinct levels of habitat quality, not 2 – 9 as the authors state.

We think that we were correct. The “random” command in NetLogo is defined in a non-intuitive way. From the NetLogo manual (<http://ccl.northwestern.edu/netlogo/docs/dict/random.html>): If “number” is positive, “random number” reports a random integer greater than or equal to 0, but strictly less than “number”.

Note 3: The authors make a great point here that designating 50 time steps as a stopping point may cause the simulation to end prematurely before IFD is reached and that this may be the reason for the hump in depicted in Figure 4e of DiNuzzo and Griffen. The authors state in their manuscript that in their model they no longer see this hump, but they do not provide a figure of this since Figure 1 of their paper does not ever adjust the personality types to be 0.9 and 0.1 rather 0.8 and 0.2.

We now show in the new Figure 2A that designating 50 time steps as a stepping point does indeed make a big difference. Referee 1 is right in pointing out that we did not show explicitly that the “hump” disappears for the 0.9-0.1 scenario. We can assure the referee that also in this scenario the time to reach the IFD decreases monotonically with the percentage of active individuals in our version of the model. Interestingly, the same happened when we applied the published NetLogo code of DiNuzzo and Griffen to this scenario. Doing the same with the other scenarios in their Figure 4 revealed that this figure is *not* based on the NetLogo code published in their Supplement. Our new Figure 2 clarifies matters and shows that (a) the hump in DiNuzzo and Griffen’s Figure 4e is an artefact, and that (b) their stopping criterion has a huge effect on model outcome.

Note 6: Here, the authors are concerned that there is an upper limit to the max feeding rate, set at 1, and that this causes a “bug” in the system where individuals choose not to move to patches with higher number of resources because they are on a patch that maxes their resource consumption limit, even though it has fewer overall resources than the richest available patch. This constitutes a major point of the authors’ submitted article, but the authors do not seem to have understood the reason behind also running the model using a Type II functional response in addition to simulations with a Type I functional response. Type I functional responses are rare in nature because they assume that individuals are never satiated and will consume whatever resources are available. However, eventually individuals’ needs are met and they stop consuming resources even when more are available in the region. Under these scenarios, there is no point for individuals to expend energy traveling to another location with more resources, potentially risking higher predation rates during travel, if they can already acquire all the resources they need in the present location. It is therefore more realistic to examine these scenarios using a type II functional response where individual feeding rate is capped and where individuals stop moving on patches that meet their needs but are not the richest in available resources. However, the increase in time to reach IFD with more active individuals is not actually so much a result of individuals taking more time to find the richest available patch because they are satiated on “suboptimal patches”, but because of changes to how patch resources are divided among individuals. Resources on a patch are no longer divided into ever smaller infinitesimal fractions as crabs are added (where a patch with 8 resources could theoretically support 4 crabs each consuming 2 resources or 400 crabs each consuming 0.02 resources, which is unrealistic), but instead resources are subtracted as crabs are added so eventually no more crabs can go on a certain patch because all resources would already be consumed (a patch with 8 resources could support 8 crabs consuming 1 resource; code lines 80 and 98 versus 89 and 107). This means that adding a crab to a patch has a much larger effect on available resources and can cause a chain reaction in additional movement as a single crab is added to a patch that had few available resources remaining. As the number of active individuals increase, so too would the probability that a chain reaction in movement would occur.

This difference in how resources are divided among population members and its effect on findings should have been highlighted better by DiNuzzo and Griffen since both calculations are valid under different ecological circumstances and produce very different results. But framing the results of Figure S1 as an error or artefact does not seem appropriate. It could however lend itself well to a discussion on ecological context and methods of modeling that context. Especially, if the authors run a sensitivity analyses varying the max feeding limit and report subsequent results. If the authors want to frame this as an error then they need to show that the results become qualitatively different when the mistakes highlighted in points 4 and 5 are corrected and should provide new model code in supplemental.

We think that a somewhat misleading formulation in our main text has contributed to a misunderstanding here. A type II functional response is naturally saturating, and therefore has an upper limit, as pointed out by the referee. In the parameterization chosen by DiNuzzo and Griffen, this upper limit is equal to 1. However, in contrast to the description given by DiNuzzo and Griffen, potential food consumption is not determined by the type II functional response, but by a different function (denoted as $L(R,C)$ in note 5 of our technical comment) that does *not* saturate. To compensate for this lack of saturation, DiNuzzo and Griffen set all consumption rates equal to 1 whenever the calculated value is larger than one. What they should have done instead is to implement the type II functional response, which automatically would have prevented unlimited consumption. The problem is further aggravated by other bugs in their code, as explained in our technical notes (4) to (6). We have expanded on these points in our manuscript and show the simulation outcomes under different implementations, the one by DiNuzzo and a corrected implementation by us (the new Figure 1; NetLogo code provided in the supplement). This now shows clearly that the finding by DiNuzzo and Griffen of increasing times until IFD with increasing proportions of active individuals is in error.

In the supplemental material authors also need to provide the code used to produce the model results depicted in Figure 1 as well as highlight the associated changes they made to the original model code of DiNuzzo and Griffen.

We reimplemented the model in NetLogo after correcting for errors, and this code is in the supplement. The C++ code of our original model is distributed over different header files and is therefore better presented as separate files in our GitHub repository, from where the code may be conveniently accessed for replication. We further have archived the submitted version of the code (16/4/2021) on Zenodo.

Point 3

Again, code for Figure 1 would best be provided in supplemental. Note that DiNuzzo and Griffen used 1000 replicate simulations rather than the 300 used here.

We have rerun the model with 1000 replicate simulations and present an updated figure (now Figure 3) in the manuscript. With regard to the C++ code, we refer the reader to our GitHub repository, which gives a more transparent overview.

Point 4

The authors are correct that time to reach IFD is an imperfect measure for quantifying such deviations in a natural system. This method is really only suitable for a highly standardized model like the one utilized by DiNuzzo and Griffen that only implements 2 personality types and runs 1000 simulations/scenario to help identify the variation in results due to differences in initial starting conditions and probability that simulation random-float numbers hit movement thresholds. The authors should focus this segment more on alternative methods to measure differences from IFD that work better for natural populations or more sophisticated models. This section would benefit greatly if the authors implement these different measurements of deviation from IFD in their own models and compare the results. Does using variance in intake rates across patches actually lead to different conclusions?

We have incorporated measurements of variance in intake rates in our simulation program. The results show, that variance drops rather fast in the beginning, and then taper out as the IFD is approached. This would be important in the case of fluctuating landscapes, as also investigated by DiNuzzo and Griffen. We have chosen to not raise this additional issue here, and therefore reduced emphasis here.

Point 5

It is true that refuge use does not necessarily translate to activity level as individuals can still be moving around to some degree while within a refuge. However, the authors seem to have misinterpreted how DiNuzzo and Griffen relate refuge use (what they call activity level) to their calculation of movement propensity. Here, refuge use was not treated as a 1-1 relationship to movement propensity but instead relied upon a prior study conducted by Belgrad and Griffen (2018) that quantified the relationship between individual refuge use and movement propensity to determine how much crabs would move in the model. The major criticism here then should not be that DiNuzzo and Griffen used the wrong study system since there is a fair amount of literature demonstrating that the species exhibits persistent individual differences in various behaviors that correlate with a number of other important characteristics. Rather the major point of concern is the number of relationships estimated in assigning population movement propensities for the model and the extra level of uncertainty built into their calculations. Instead of using movement propensities measured from a large sample of the population, DiNuzzo and Griffen estimate individual refuge use propensity from a size distribution of the population and then use that estimation to estimate movement propensity of the population. The authors should instead focus their critique on this level of uncertainty or that DiNuzzo and Griffen needed to use a study system where personality distribution did not have to be estimated.

We have revisited our critique with regards to the mentioned aspects and no longer focus on the somewhat uncertain relation between refuge usage and activity level.

Figures 3 and 4 provide graphs of how the degree of personality variation can affect the time to reach IFD from theoretical populations. The purpose of Figure 5, does not appear to really be about illustrating how variation in personality can affect the time to IFD which they do with their theoretical populations. Figure 5 is a rough calculation of how much longer it should take a natural population to reach IFD given that individuals do not follow "ideal" movement probabilities. If one were to vary the population personality distributions for such a calculation, it would no longer be a real-world personality distribution. Is it natural for 90% of individuals to have an activity level of 0.9? It would have been better if the DiNuzzo and Griffen paper used multiple populations to provide that information, but that also seems to be the point of the first part of their manuscript. Studies almost never report the personality distribution of populations and there is a lack of data on this subject. Still, they did find 10 studies that do seem to report that information so they could have potentially used that data to improve their estimates further.

The referee makes valid points here, and we have clarified our critique in response. The issue with the rough calculation is that as far as we can see, it does not take the observed variation into account, and therefore ignores the role of personality variation completely.

Response to Referee 2:

This manuscript clarifies and enriches some potentially key processes first outlined by DiNuzzo and Griffen (2020) influencing the time required for hypothetical populations to converge on an ideal free distribution and the commonly-observed tendency for under-matching to occur as a result. The authors central thesis is that any delays in decision-making will automatically extend the time to equilibrium by definition. This seems a subtle, but important, point since other many factors (e.g. variation among individuals due to differences due to internal motivation, physiological state, perception, or cognition) could act in this way in addition to variation in personality syndrome.

The authors also clarify that a key component in any such model is whether decisions are made sequentially by individuals across the population or simultaneously. Changing this assumption changed several outcomes in important ways. Such considerations about discrete vs continuous representations of time can fundamentally alter outcomes in many models in ecology, including the most basic of density-dependent formulations. Given that we can rarely (perhaps never) be sure that either continuous or discrete formulations are truly correct, it is important to recognize this as a key biological constraint on all model formulations. The authors provision of code for IDF formulation according to simultaneous decision-making is a most useful contribution to the field.

We are pleased to hear that Referee 2 appreciates the main goal of our contribution.

I am less enthusiastic about the authors' complaint that DiNuzzo and Griffen's (2020) crab population represents a less than perfect example of the processes discussed in the theoretical part of the paper. This is always true, of course, but particularly so for IDF papers. Theoreticians gravitate to IDF models precisely because of their tractability and simplicity of assumptions. Even slight departures from the most simple underlying assumptions can dramatically complicate model prediction, as my lab knows from direct experience (Avgar et al. 2021). It is unworthy to complain that nature doesn't comply with our theoretical preference for simplicity. Provision of empirical examples, even those with potentially serious flaws, help readers recognize ways that important concepts might be useful in better understanding their own study systems and encourage useful follow-up studies. They should rarely be treated as logical "proofs" per se.

It was not our intention to criticize the empirical example for its noncompliance with the assumptions of a highly simplified model. Most of our own work is on applying relatively simple concepts to empirical systems with the goal of achieving enlightenment while being aware of the discrepancy between simple models and natural systems. In response to the above comments, we have substantially modified our critique of DiNuzzo and Griffen's empirical example, carefully avoiding the (wrong) impression that our critique is targeted against the fact that their mud crab system does not fully comply with their model. We no longer focus our critique on the underlying data (except for the fact that they are derived from a predation cue treatment), and instead focus on the treatment of data to arrive at figure 5, a treatment which in our view ignores the very role of variation it claims to demonstrate.

Response to Referee 3:

In this manuscript, the authors comment on a recently published article by DiNuzzo & Griffen (hereafter D&G, for concision) in Proceedings B entitled "The effects of animal personality on the ideal free distribution". I am pleased to see this comment come through. It is timely and, I think, an important contribution to this discussion. Because of the brevity of the comment, I will address each of the authors' main 5 points (which, I think, should be better identified instead of using simple parenthetical numbers).

We are pleased about this positive evaluation, and we have now replaced the parenthetical numbers by meaningful descriptors. Thanks for this suggestion!

1. Interpretation of the results in terms of personality variation

I think the authors are spot-on in their first point. Because all animals in the D&G simulation have perfect knowledge of the landscape, and because some animals move more slowly, the IFD is achieved but slowly. This is, as they say, due to inefficient movement, and this point is reflected across several of my reviews of D&G's original work. If we accept that variation in movement rates constitutes

"personality" (a common assertion among wildlife ecologists, for example, for better or worse), then this is not necessarily the problem the authors assert. Still, their observations are correct, leading D&G's model to produce what I have referred to as a "self-fulfilling prophecy". This seems obvious to me, but many readers may not pick up on this, and so here the comment is providing an important observation about D&G's work.

Obviously, we fully agree.

However, I think their second paragraph in this section is in some ways seeking a fight where one is not necessary. D&G's work was not intended to discuss the eco-evolutionary causes of personality and the IFD, but rather to address undermatching and the role of variation in personality (there expressed as variation in movement rates) in shaping the rate at which an IFD is achieved. Nothing the authors say in that paragraph is technically wrong, but their points could not be adequately addressed in the D&G model because stable coexistence of different personalities would require a dedicated game-theoretic-styled model wherein different personalities (there, movement strategies) could be evaluated for their individual fitness. I agree with the authors implication here that this would be a valuable contribution, but criticizing the D&G study for omitting something like this is inappropriate as it was outside the scope of the original work.

Referees 1 and 3 have convinced us that our remarks on the evolutionary consistency of ecological models should be decoupled from our critique of the DiNuzzo and Griffen paper. Still, we think that the points we make are worth raising. Therefore, we have shifted this section to the end of our manuscript, where it now gives more of a general outlook

2. Deficiencies in the technical implementation of the model

I really appreciate the reference to a Zenodo page laying out all code and highlighting errors in the code and thank the authors for providing this.

In response to Referee 1, we now make our technical comments on the code even more accessible by including them in a supplement. Moreover, we now also include corrected versions of the NetLogo code. The C++ code of our own simulation remains on Zenodo.

3. The effects of population size on deviations from the IFD

This is a good catch, and an elegant way to demonstrate the problems caused by treating the population size problem in the manner of D&G given how their algorithm works. I suspect that one could also approximately re-create the authors' Figure 1 using the original outputs of the D&G work by simply dividing the time to achieve the IFD by the number of individuals in the population; if so, that would be an even more powerful demonstration.

This is a very good suggestion. We have made attempts in this direction, but abandoned these because the errors in DiNuzzo & Griffen's code prevent such a direct comparison. In addition, DiNuzzo and Griffen essentially convolve several geometric distributions, whereas our simulations allow individuals to move in parallel and therefore calculate the maximum value of several draws from two different exponential distributions. These outlined calculations ignore however the saturation effect on the landscape, which is much more difficult to capture analytically.

Additionally, their note that when individuals occupy patches simultaneously they benefit by staying if others leave is well-established. There is a fair amount of literature demonstrating this implicitly by virtue of the mathematics at play (the various works of Cantrell, Cressman, and Cosner come to mind).

We cited a paper from our own group where this is demonstrated (reference [4] in the revised version), but we agree that this effect is implicit in many models of interference competition. We would appreciate if the referee provide us with additional references in this direction.

4. The measure used for quantifying deviations from the IFD

Again, there are some good observations here that are driven by the simplicity of the original D&G model. The authors are correct that time-to-IFD is not a flawless metric given how individual discrepancies from an optimal movement strategy and initial distributions of the population can affect emergence of the IFD. However, their third point that time-to-IFD is only sensible when an IFD can be achieved seems somewhat unnecessary to me, since that is the condition imposed by the original D&G model. In this third point, the authors are in fact arguing not for a different metric of IFD (e.g. variation in intake rates), but rather for a fully different parameterization of the model and its rules. I would suggest this point be revised to make this more explicit because, frankly, I think the authors are wrong to suggest that variation in intake rates is superior to time-to-IFD in this instance. Yes, at IFD intakes rates will be identical across patches and individuals, but the spatial consequence of that is more animals in high quality patches, which is how D&G represented their original work (as I recall).

Still, if you change the way the model works by including e.g. seasonality in patch quality, then yes, a different metric as suggested by the authors could be much more appropriate. Hence my suggestion that this section be revised to reflect model structure more explicitly, and particularly how model structure affects what metrics are most appropriate.

We fully agree with the referee: under highly idealize conditions (where the IFD will indeed be reached in finite time), time-to-IFD may be a reasonable measure. However, DiNuzzo and Griffen also consider a scenario where the environment is constantly changing. Under this scenario, time-to-IFD does not make much sense (in particular in case of rapid fluctuations), and we do not understand how DiNuzzo and Griffen could arrive at their Figure S2.

5. The relevance of the empirical data set

Excellent points. My only suggestion here is that the authors might make reference to other empirical studies highlighting the difficulty in evaluating IFD in real animals due to the various problems identified by the authors here.

This is a good suggestion, but we have moved our focus here on the theoretical issues with the data analysis in D&G, also in view of the advice of Referee 2.

Please add line numbers so that we can pointedly address specific portions of the manuscript more efficiently.

We have done so. Thanks again for your insightful comments.

Appendix B

Details matter when modelling the effects of animal personality on the spatial distribution of foragers

Christoph Netz, Aparajitha Ramesh, Jakob Gismann, Pratik R. Gupte, Franz J. Weissing

Response to Referees

We are pleased to hear that the Associate Editor as well as Referee 1 view our extensive changes to the original manuscript as a major improvement, and we thank them again for their efforts regarding our manuscript. Here, we address the two remaining concerns of Referee 1.

Lines 24-28: This critique about D+G paper seems to be missing the point about the original manuscript and should still be reduced. The D+G paper is not focusing so much on variation in personality types as Netz et al. describe, but on the existence of personalities itself (where individuals are not completely flexible behaviorally in being able to respond to their environment). The D+G paper is demonstrating that the presence of a personality, particularly an inactive personality, “may inhibit the ability of a population to track changes in habitat quality”. The point is that a personality could change the time to reach IFD at all and that it is important to know the variation in personalities within a population because this will help determine the time it takes to reach IFD. The D+G paper is not at all about personality variation itself increasing time to IFD as Netz et al. seem to focus upon. However, a critique could be added that D+G focus on how that inflexible behavior slows time to matching and that there may be mechanisms where organisms can communicate or observe each other to speed the approach to IDF although that would involve a completely different spectrum of personality. If one wants to argue this, then the authors should elaborate their critique to explain how D+G focus only on activity level as their representative of personality and that other behavioral facets such as sociability and numbers of leaders or communicators may speed the approach to IDF (with examples of systems where this may occur).

We partly agree with the referee that D+G tend to view personality types as examples of inflexible and inefficient behaviour. However, D+G go beyond treating the simple existence of animal personalities and their effect on the IFD. Repeatedly, the importance of the personality *distribution* is stressed in their paper, not just the existence of personality itself.

To clarify this issue, we have now dropped the sentence in lines 26-28: *“Understanding this reason underlying DiNuzzo and Griffen’s results is important, as their study might lead to the wrong conclusion that, quite generally, variation in animal personality types slows down the approach to the IFD.”* Instead, we added the following opening sentences to the “Outlook” section of our comment: *“We have the impression that DiNuzzo and Griffen view “personalities” mainly as (maladaptive) deviations from optimal or efficient behaviour. In contrast, many studies show that personality variation is often shaped by adaptive evolution [7-13]. For example, Wolf and colleagues [4] demonstrate that “inactivity” (called “unresponsiveness in [4]) may be viewed as an efficient strategy in achieving a high foraging success and approaching an ideal free distribution. An adaptive perspective on personality variation leads to novel eco-evolutionary questions regarding the interplay of individual behavioural variation and the spatial distribution of foragers.”*

Lines 149-163: Again Netz et al. are focusing on the idea that D+G are using Figure 5 to demonstrate the role of personality variation on time to reach the IFD. This is not the case since they already illustrate how personality variation can affect time to IFD with their earlier simulations of artificial

personality distributions (e.g. D+G Figure 3). Figure 5 is used to show how much longer it should take for a natural population with a “known” personality distribution (calculated from size) to reach IFD over a theoretical population that does not have personalities and where each individual is perfectly flexible in their behavior and active. The calculation takes the observed variation into account by looking at the variation in body size within the population, transforming that into personality from previously made regressions of body size and personality (so x individuals have personality 0.01, y individuals have 0.014, z individuals have 0.2.... etc.) then running the simulation to see how long that population takes to reach IFD with that natural personality distribution. It’s a real-world example of how much time inflexible behavior in a natural population may add to reaching IDF and a demonstration of how their model or a model built for a specific study system could be applied to other populations. It would have been nice if D+G did this with more populations as comparison or looked at subpopulations, but was not necessary to prove their point.

Again, we partly agree with the referee. In response to the above comments, we have now dropped the final sentences (lines 157-163) of this paragraph. However, we disagree with the referee on two issues. First, D+G did not run their simulation with the personality data obtained from the mud crab system, as becomes evident from the last sentence of their Methods section:

Therefore, the ratio of this ideal free probability of leaving a patch and the estimated dispersion probability for our 264 crabs yields the relative increase in time to reach the IFD across oyster reefs for the crabs in each of our 10 000 replicate calculations for this population.

Second, this calculation, although it is not explained in sufficient detail, does not take into account the distribution of personalities, as this would either require a) usage of the simulation program, or b) the convolution of several geometric distributions, from which the distribution of movement times of individuals could be derived (if we follow the logic of D+G’s modelling approach). This would certainly have been mentioned in their article.